# CDFFusion: A Color-Deviation-Free Fusion Network for Nighttime Infrared and Visible Images

**DOI:** 10.3390/s25237337

**Published:** 2025-12-02

**Authors:** Hao Chen, Tinghua Zhang, Shijie Zhai, Xiaoyun Tong, Rui Zhu

**Affiliations:** Department of Electronic and Optical Engineering, Space Engineering University, Beijing 101416, China; 18317921185@163.com (H.C.); 15810887677@163.com (S.Z.); xiaoyun_t@hgd.edu.cn (X.T.); zoranr@163.com (R.Z.)

**Keywords:** night scene, image enhancement, image fusion

## Abstract

The purpose of infrared and visible image fusion is to integrate their complementary information into a single image, thereby increasing the amount of information expression. However, previously used methods often struggle to extract information hidden in darkness, and existing methods—which integrate brightness enhancement and image fusion—can cause overexposure, image blocking effects, and color deviation. Therefore, we propose a visible light and infrared image fusion method, CDFFusion, for low-light scenarios. The premise is to utilize Retinex theory to decompose the illumination and reflection components of visible light images at the feature level before fusing and decoding the reflection features with infrared features to obtain the Y component of the fused image. Next, the proposed color mapping formula is used to adjust the Cb and Cr components of the original visible light image; finally, the Y component of the fused image is concatenated to obtain the final fused image. The SF, CC, Nabf, Qabf, SCD, MS-SSIM, and Δ*E* indicators of this method reached 17.6531, 0.6619, 0.1075, 0.4279, 1.2760, 0.8335, and 0.0706, respectively, on the LLVIP dataset. The experimental results show that this method can effectively alleviate visual overexposure and image blocking effects, and it has the smallest color deviation.

## 1. Introduction

Visible images usually contain rich texture detail information, but they are prone to the loss of target information in complex scenes; meanwhile, infrared images—which are formed based on thermal radiation information—are not easily affected by harsh conditions but lack detailed descriptions of the scene. Therefore, infrared and visible image fusion (IVIF) can make full use of their complementary information and significantly improve the comprehensive perception ability of the scene. However, with current approaches, most of the texture details in visible light images captured in low-light environments with low visibility are obscured. The general approach to addressing this issue is to first preprocess the visible light image using a low-light enhancement method and then fuse it with the infrared image. However, this can cause color distortion in some areas; therefore, the matter of how to organically combine low-light image enhancement with IVIF is a significant challenge.

Figure 1 shows a visible image, the corresponding infrared image, the result processed by RFN-Net, and the result processed using LEDNet [1] preprocessing followed by RFN-Net. First, previous IVIF methods fail to extract the information of visible images obscured at night (as shown in the red box in Figure 1c). In contrast, fusing the visible image after enhancement preprocessing causes color distortion in some areas (as shown in the green box in Figure 1d, where white zebra crossings are rendered green).

Second, existing nighttime IVIF methods, while solving the above two problems, introduce new issues. Figure 2 displays a visible image, the corresponding infrared image, and results obtained using DIVFusion, LEFuse, LENFusion, and proposed method. It can be observed that DIVFusion and LEFuse overemphasize the difference between high and low gray values and focus on highlighting high-gray-value regions, which has two consequences: first, overexposure occurs in certain parts of the image (the lower left corners in Figure 2c,d); second, severe block artifacts (the mosaic effect, as shown in the red boxes in Figure 2c,d) or false edges appear in non-edge or weak-edge regions. Although LENFusion avoids overexposure, it differs from the two methods previously discussed by focusing on suppressing low-gray-value regions, which also leads to two problems: in some weak-edge regions, the low-gray-value parts are severely distorted, leading to information loss (as shown in the red box in Figure 2e); in other weak-edge regions, the colors of these low-gray-value parts are rendered extremely dark, causing unnatural false edges. Additionally, these methods only process the Y channel of visible images without handling the Cb and Cr channels, which changes the hue and saturation of the source images, resulting in varying degrees of color deviation.

To solve the above problems, this study proposes a two-stage network for joint low-light image enhancement and image fusion without color deviation. It can mitigate visual overexposure, image blocking artifacts, and color deviation. First, an encoder with a Feature Pyramid Network (FPN) structure is used to extract deep features of visible and infrared images. Then, combined with RetiNex theory, a decomposition network is designed at the feature level to separate the illumination component and reflectance component of the visible image. Next, the proposed formula is applied to process the Cb and Cr components of the original visible image. The reflectance component features and infrared features are concatenated and input into the fusion network to obtain the Y component of the fused image. Finally, the Y component is concatenated with the processed Cb’ and Cr’ components of the visible image to generate the final fused image.

In summary, the main contributions of this study are as follows:It proposes CDFFusion, a two-stage network for joint low-light image enhancement and image fusion, which can mitigate visual overexposure, image blocking artifacts and color deviation;A brightness enhancement formula without color deviation is proposed, which processes the three components (Y, Cb, Cr) simultaneously, and the processed results have the smallest color deviation.

## 2. Related Work

Deep-learning-based visible and infrared image fusion methods can be roughly divided into three categories: those based on Convolutional Neural Networks (CNNs), those based on Autoencoders (AEs), and those based on Generative Adversarial Networks (GANs). Ma et al. clearly defined the required information for IVIF as the significant thermal targets of infrared images and the background texture structures from visible images. On this basis, they proposed STDFusionNet [2], a fusion method based on significant target detection. It uses the significant target mask to label the more important areas in the infrared image and combines a specific loss function to provide guidance for the final fusion. As an end-to-end model, it can implicitly complete significant target detection and key information fusion, thereby retaining the target prominent information in the source image.

Zhang et al. proposed a Squeeze-and-Decomposition Network, SD-Net [3], which models image fusion as the extraction and reconstruction of gradient and intensity information. As different fusion tasks share a similar goal—fusing images by integrating important and complementary information from multiple source images—Xu et al. proposed a unified unsupervised end-to-end image fusion network, U2Fusion [4], which is characterized by its ability to perform information measurement on extracted features to automatically estimate the importance of source images. Zhao et al. proposed a method called CDDFuse [5], which features a correlation-driven dual-branch feature decomposition as its core. It first extracts cross-modal shallow features through Restormer blocks and then decomposes the features using a dual-branch Transformer–CNN feature extractor that contains Lite Transformer (LT) blocks and Invertible Neural Network (INN) blocks. Meanwhile, a correlation-driven loss is introduced to ensure that low-frequency features are correlated and high-frequency features are uncorrelated. Finally, the fused image is output through LT global fusion and INN local fusion layers.

Some studies have tackled the IVIF task from other perspectives, such as light perception and multi-resolution. Tang et al. proposed a progressive image fusion network based on light perception (PIAFusion) [6] to address the issue of light’s impact on image fusion. They designed a light perception sub-network to estimate the light distribution of images and calculate the light probability. Thus, the network can adaptively perceive the intensity distribution of images and utilize the light probability to construct a light perception loss function to guide the training process of the fusion network. The fusion results can better cope with the influence of light changes.

Li et al. observed that only the output of the last layer is used as image features in traditional CNN-based fusion methods, which leads to the loss of a lot of useful information obtained from intermediate layers. Therefore, they were the first to introduce the DenseBlock into AE and proposed DenseFuse [7], where the output of each layer is directly passed to all subsequent layers, thus fully utilizing the information from intermediate layers and enhancing the network’s feature extraction capability. The encoder consists of convolutional layers and DenseBlocks for extracting deep features; the fusion layer employs an additive strategy and l1-norm strategy to fuse features; the decoder contains four convolutional layers for reconstructing the fused image. Although this approach can more fully transfer and reuse feature information, the manually designed fusion strategy may limit the performance of the final result. Later, Li et al. successively proposed the network framework NestFuse [8], based on nested connections and spatial/channel attention models, as well as a novel end-to-end fusion network architecture (RFN-Nest) [9]. The former adds upsampling operations and nested connections between network layers, enabling more thorough utilization of deep features for image reconstruction during the decoding stage. The latter builds upon this by adopting a Residual Fusion Network (RFN), where the encoder extracts multi-scale deep features through max pooling, and the RFN, composed of several convolutional layers, is trained with a new loss function to achieve a learnable fusion strategy rather than a manually designed rule. Xu et al. proposed a classification saliency-based pixel-level fusion method, CSF [10], which classifies different source images, uses the classification results to represent the saliency of each pixel in the source images, and finally fuses the feature maps using this saliency to generate the fusion result.

Ma et al. observed that different modal source images (such as infrared and visible light images) may not require the same transformation or representation, and the measurement of activity levels and fusion rules are mostly manually designed; these are complex and have high implementation difficulty and computational cost. Ma et al. first proposed an end-to-end model based on a Generative Adversarial Network (GAN) and called it FusionGAN [11]. It represents the fusion problem as an adversarial problem, while avoiding the manual design of complex activity level measurement and fusion rules; later, the team noticed that most of the richness in maintaining the inherent information of infrared and visible light images struggles to achieve a good balance. For example, some infrared images may also contain more detailed texture than visible light images, and vice versa: visible light images may have higher contrast than infrared images. To address this issue, they designed a Generative Adversarial Network with Multiclassification Constraints (GANMcC) [12] with multiple classification constraints, which represents the fusion problem as the simultaneous estimation of multiple distributions.

To address the shortcomings of a single discriminator, Zhou et al. [13] developed a learning architecture with two discriminators to ensure that the generator can capture comprehensive features. Xu et al. designed a conditional generative adversarial network with dual discriminators (DDcGAN) [14]. This improved the traditional GAN model to preserve the features of images from two different modalities. The dual discriminators use the gradients of the source images as real data, avoiding the imbalance problem that a single discriminator might cause, but simultaneously increasing the complexity of network training.

DIVFusion [15] was the first to combine IVIF and low-light enhancement. It decomposes the visible image into reflectance features and illumination features at the feature level; then, it enhances and fuses the reflectance features with infrared features. LENFusion [16] adopts the idea of pre-enhancement, re-enhancement, and fusion. It draws on the idea of PIAFusion by pre-training a binary classifier and using its classification results as part of the loss function of the backbone network. LEFuse [17] introduces a hybrid module of Transformer and CNN and designs the overall network structure into a symmetric structure similar to the U-net style.

## 3. Methods

The method proposed here improves nighttime visibility, retains the complementary information of the two modalities, alleviates overexposure and block artifacts, and eliminates color deviation. This section details the two sub-networks of the entire framework, including the network structure and loss function.

### 3.1. Overall Framework

As shown in Figure 3, let $I_{vi}^{Y}$ and $I_{ir}$ represent the Y channel of the visible image and the infrared image, respectively. After passing through the decomposition network and fusion network, the Y channel of the fused image is obtained. Then, it is concatenated with the processed Cb and Cr channels of the visible image to obtain the final fusion result. The entire process is divided into two stages: feature extraction and image fusion.

### 3.2. Reflectance–Illumination Decomposition Network (RID-Net)

The specific structure of the reflectance–illumination decomposition network is shown in Figure 4, which consists of two parts: a visible image reconstruction network and an infrared image reconstruction network. The former is used to separate the reflectance and illumination components of the visible image at the feature level and reconstruct them into images, while the latter is used to extract deep features of the infrared image and reconstruct them into images. First, in the feature extraction stage, the Y channel of the visible image is input into the encoder with an FPN structure to extract deep features ϕ. The specific structure of the encoder is shown in Figure 5. The original image passes through a series of convolutional layers with Sobel operators and residual connections, resulting in four feature maps of different sizes. In the last three of these structures, the stride of the convolutional layers is set to 2 to achieve downsampling. Then, 4 1 × 1 convolutional layers are used to unify the number of channels of the feature maps obtained just now, which is set to 256 here. After that, these processed feature maps are upsampled by 2 times and summed to complete multi-scale fusion. This process can be expressed as:(1)$$\phi =E(I_{vi}^{Y})$$

In the feature separation stage, according to RetiNex theory, an image can be decomposed into the product of the reflectance component (R) and the illumination component (L) (as shown in the following formula). Therefore, two Efficient Channel Attention (ECA) modules are used to separate the reflectance features $\phi_{R}$ and illumination features $\phi_{L}$ from them.(2)I=R×L

The structure of the ECA module is shown in Figure 6. First, global average pooling is performed on each channel of the input feature map to obtain the global feature of each channel. Then, one-dimensional convolution and activation are applied to these global features to obtain attention weights. The input feature map is reweighted using these weights to obtain the output feature. Here, GAP and σ represent global average pooling and the sigmoid activation function, respectively. Compared with other attention mechanism modules, such as Squeeze-and-Excitation Networks (SE) and the Convolutional Block Attention Module (CBAM), the ECA module avoids fully connected layers, significantly reducing the number of parameters, and uses 1D convolution instead of 2D convolution for dimensionality reduction operations, preserving complete channel information. This process can be expressed as:(3)$$\begin{array}{l} ECA_{R}(\phi )=\phi_{R}\\ ECA_{L}(\phi )=\phi_{L} \end{array}$$

In the final decoding stage, they are input into two decoders with identical structures, and the reflectance component $I_{R}$ and illumination component $I_{L}$ are obtained through reconstruction. Both decoders are composed of four stacked groups of convolutional and activation layers. The first three activation layers use the LRelu function, and the last one uses the sigmoid function. This process can be expressed as:(4)$$\begin{array}{l} D_{R}(\phi_{R})=I_{R}\\ D_{L}(\phi_{L})=I_{L} \end{array}$$It should be noted that the decoding part here is only for better image reconstruction, and does not participate in the operation of the next stage.

Similarly, the infrared reconstruction network adopts a basically consistent structure, which is not further discussed here.

The loss function of this part of the network is as follows:(5)$$L_{1}=\lambda_{1}L_{rev}+\lambda_{2}L_{per}$$The loss function $L_{1}$ of the visible image reconstruction network consists of two parts: reconstruction loss $L_{rev}$ and perceptual loss $L_{per}$. The reconstruction loss $L_{rev}$ is derived from Formula (2), which is expressed as follows:(6)$$L_{rev}={\left\|I_{R}\times I_{L}-I_{vi}^{Y}\right\|}_{1}$$

On this basis, we predict that the brightness-enhanced result (i.e., the reflectance component $I_{R}$) and the result of the original image $I_{vi}^{Y}$ after histogram equalization will have as similar representations as possible in the feature domain of the VGG-19 network. Thus, the perceptual loss $L_{per}$ is defined as:(7)$$L_{per}={\left\|VGG(I_{R})-VGG[hist(I_{vi}^{Y})]\right\|}_{2}^{2}$$
where “hist” represents the histogram equalization operation. Here, the Conv4-1 feature of the VGG-19 network is selected.

The loss function $L_{2}$ of the infrared reconstruction network consists of reconstruction loss $L_{rei}$ and structural similarity loss $L_{s1}$, which can reconstruct results similar to the original image in terms of both intensity and structure. It is defined as follows:(8)$$\begin{array}{l} L_{2}=\lambda_{3}L_{rei}+\lambda_{4}L_{s1}\\ L_{rei}={\left\|{I_{ir}}^{\prime }-I_{ir}\right\|}_{1}\\ L_{s1}=1-ssim({I_{ir}}^{\prime },I_{ir}) \end{array}$$Here, $\lambda_{1}∼\lambda_{4}$ are weight hyperparameters used to balance the various parts of the loss.

### 3.3. Fusion Network

In the fusion stage, the reflectance features $\phi_{R}$ of the visible image and the infrared features $\phi_{ir}$ are concatenated at the channel level and input into the fusion network. Consistently with the decoder structure in the previous stage, it is also composed of 4 stacked groups of convolutional and activation layers. At this time, its output $I_{f}^{Y}$ is taken as the Y channel of the fused image, which is used to participate in the final channel fusion.

According to the **ITU-R BT.601** international standard [18], the conversion formula from the RGB space to the YCbCr space of an image is:(9)$$\begin{array}{l} Y=0.299R+0.587G+0.114B\\ Cb=128-0.168736R-0.331264G+0.5B\\ Cr=128+0.5R-0.418688G-0.081312B \end{array}$$

In the HSI color space, the conversion formula from RGB to HSI is:(10)$$\begin{array}{l} H=\left\{\begin{array}{c} \theta ,B\leq G\\ 360^{\circ }-\theta ,B>G \end{array}\right.\\ \theta =\arccos\left\{\frac{\frac{1}{2}[(R-G)+(R-B)]}{\left[{\left(R-G\right)}^{2}+(R-B){\left(G-B\right)}^{1/2}\right]}\right\}\\ S=1-\frac{3}{\left(R+G+B\right)}[\min(R,G,B)] \end{array}$$
where H represents hue and S represents saturation. From the above formula, it can be proven that only when the three components R, G, and B are scaled proportionally are the hue and saturation of the pixel unchanged. Substituting the result into Formula (9), it can be deduced that [refer to Appendix A]: only when (Cb − 0.5) and (Cr − 0.5) are scaled synchronously with Y, R, G, and B are scaled proportionally (the pixel value range has been normalized to [0, 1]), and the hue and saturation remain unchanged. Thus, we naturally derive the mapping formula for the Cb and Cr channels of the visible image:(11)$$\begin{array}{l} scale=I_{R}/I_{vi}^{Y}\\ {I_{vi}^{Cb}}^{\prime }-0.5=(I_{vi}^{Cb}-0.5)\times scale\\ {I_{vi}^{Cr}}^{\prime }-0.5=(I_{vi}^{Cr}-0.5)\times scale \end{array}$$
where “scale” is the brightness gain image, while $I_{vi}^{Cb},I_{vi}^{Cr},{I_{vi}^{Cb}}^{\prime },{I_{vi}^{Cr}}^{\prime }$ correspond to the Cb channel and Cr channel of the original visible light image, as well as the Cb channel and Cr channel of the adjusted visible light image. Unlike other methods that use loss functions for weak constraints, the above formula applies strong constraints on the proportion between the three components R, G, and B of the visible image, thus avoiding color distortion. Finally, $I_{f}^{Y}$ is concatenated with ${I_{vi}^{Cb}}^{\prime }$ and ${I_{vi}^{Cr}}^{\prime }$, and converted back to the RGB space to obtain the final fused image $I_{f}$.

The loss function $L_{3}$ of the fusion network is as follows:(12)$$L_{3}=\lambda_{5}L_{vi}+\lambda_{6}L_{ir}+\lambda_{7}L_{aux}+\lambda_{8}L_{grad}+\lambda_{9}L_{s2}$$It consists of visible light intensity loss $L_{vi}$, infrared intensity loss $L_{ir}$, auxiliary intensity loss $L_{aux}$, gradient loss $L_{grad}$, and structural similarity loss $L_{s2}$, which are defined as follows:(13)$$\begin{array}{l} L_{vi}={\left\|I_{f}^{Y}-I_{R}\right\|}_{1}\\ L_{ir}={\left\|I_{f}^{Y}-I_{ir}\right\|}_{1}\\ L_{aux}={\left\|I_{f}^{Y}-\max(I_{R},I_{ir})\right\|}_{1}\\ L_{grad}=\left\|\nabla I_{f}^{Y}-\max(\nabla I_{R},\nabla I_{ir})\right\|\\ L_{s2}=1-\frac{1}{2}[ssim(I_{f}^{Y},I_{R})+ssim(I_{f}^{Y},I_{ir})] \end{array}$$
where ∇ represents the gradient operation, and the Sobel operator is used. These losses can force the fused image to retain more prominent intensity and gradient information from the source images and maintain a relatively consistent structural similarity with the source images. Here, $\lambda_{5}∼\lambda_{9}$ are weight hyperparameters used to balance the various parts of the loss.

## 4. Experiments

### 4.1. Experimental Configuration

To comprehensively evaluate the proposed method, extensive experiments are conducted on the LLVIP dataset [19]. The LLVIP dataset is a paired visible–infrared dataset for low-light scenarios, containing 33,672 images (16,836 pairs). Among them, 240 pairs of infrared and visible images are selected for the training phase, and 50 pairs are selected for the testing phase. These images have been strictly registered. The results of this study are compared with five fusion methods, including one AE-based method (RFN-Nest), one GAN-based method (FusionGAN), and three of the latest nighttime IVIF methods (DIVFusion, LEFuse, and LENFusion). The implementation of all methods is based on publicly available code.

Six metrics are used in the quantitative evaluation phase, including one image feature-based metric (Spatial Frequency, SF), one structural similarity-based metric (Multi-Scale Structural Similarity, MS-SSIM), and four metrics based on the source images and generated images: Correlation Coefficient, CC; Sum of Correlation Differences, SCD; Gradient-Based Fusion Performance, Qabf; and Fusion performance based on artifact detection (Nabf). Here, a smaller Nabf value indicates a better fusion effect.

In addition, in order to comprehensively and objectively evaluate the differences in hue and saturation between two pixels and verify the validity of the proposed color mapping formula, a color deviation index ΔE is now proposed, which is expressed as follows:(14)$$\begin{array}{l} \Delta E=\sqrt{{\left[\bar{S}(H_{1}-H_{2})\right]}^{2}+{\left(S_{1}-S_{2}\right)}^{2}}\\ \bar{S}=\frac{1}{2}(S_{1}+S_{2}) \end{array}$$

Here, $H_{1},H_{2},S_{1},S_{2}$ have been normalized. Considering that, when the saturation S is very small, the color is very close to gray, it is meaningless to discuss the hue difference at this time. Therefore, a coefficient $\bar{S}$ is added before the hue difference $\left(H_{1}-H_{2}\right)$. It is worth noting that this formula is used to measure the color difference between two pixels. However, when measuring the color difference between two images, it is necessary to apply this formula pixel by pixel to both images and calculate the average value. This formula shows that, the larger the value of ΔE, the greater the deviation in hue and saturation between the two images.

In the training phase, the 240 pairs of images used for training are randomly cropped to a size of 224 × 224 pixels, and the batch size is set to 5. All the parameters of the networks are updated via the Adam optimizer. The initial learning rates of the visible and infrared images in the decomposition network are 10^−4^ and 10^−3^, respectively, and they are decayed to 0.1 and 0.01 of the initial values after 50 and 75 epochs. The fusion network uses a fixed learning rate set to 2 × 10^−5^. In addition, the weight hyperparameters of the loss functions of the decomposition network and fusion network are set as follows:(15)$$\lambda_{1}=10,\;\lambda_{2}=5,\;\lambda_{3}=10,\;\lambda_{4}=1,\;\lambda_{5}=2,\;\lambda_{6}=2,\;\lambda_{7}=6,\;\lambda_{8}=10,\;\lambda_{9}=30$$

The entire network was trained on an NVIDIA GeForce RTX 3060 GPU and an 11th Gen Intel(R) Core(TM) i5-11400F @ 2.60GHz, 2592 MHz CPU using the PyTorch-2.3.1 framework. The training time was 4744.13 s. The testing times on the LLVIP and M3FD datasets were 511.84 s and 61.22 s, respectively.

### 4.2. Results and Analysis

#### 4.2.1. Qualitative Comparison

Figure 7 shows the visualization results of different fusion methods. As mentioned earlier, previous fusion methods fail to extract the information of visible images obscured at night. In the results of RFN-Nest and FusionGAN, the upper part of the images in the first row is still completely black, with no significant improvement in brightness, and the zebra crossings in the red boxes become blurred, with those in the upper right corner showing a greenish tint. In the second row of images, most details of the road in the boxes are lost, and the headlight area in the lower right corner is covered with a shadow. In the third row of images, the road details are completely lost. For the results of DIVFusion, LEFuse, and LENFusion, the color saturation in Figure 7e,f is relatively low, leading to an overexposed feeling in some areas (as shown in the green boxes in the second row of images). In Figure 7g, areas such as zebra crossings are prone to distortion, resulting in information loss (as shown in the red boxes in the first and second rows of images). In areas such as damaged roads, the color is darkened, forming unnatural false edges (as shown in the green box in the first row of images). In addition, these three methods all cause block artifacts (as shown in the boxes in the third row of images). In the third row of images, the color saturation of buildings and leaves is significantly low, with varying degrees of color deviation. In contrast, the proposed method alleviates all the above issues and achieves a better visual effect without color deviation.

#### 4.2.2. Quantitative Comparison

Table 1 presents the average quality metrics of different fusion methods on the LLVIP dataset. The method proposed in this study ranks first in the Qabf, MS-SSIM, and ΔE metrics; second in the CC and SCD metrics; and third in the SF and Nabf metrics. The optimal Qabf metric indicates that this method can suppress the image block effect while better preserving the high-frequency details. The optimal MS-SSIM metric shows that it can better maintain the consistency of the fused image with the source image. The optimal deltaE metric indicates that this method can better maintain the color consistency of the fused image with the source image. The suboptimal CC and SCD metrics show that this method can better maintain the linear correlation between the fused image and the source image. The SF and Nabf metrics are not optimal, which is understandable because this method focuses on the consistency of the structure, detail features, and color distribution of the fused image with the source image rather than excessive sharpness. This method scores higher than previous methods in the Nabf metric and is superior to other current methods because previous methods cannot effectively preserve the information of the source image such as high-frequency details, so they hardly introduce excessive artifacts. The superiority of this method over other current methods also indicates that this method can preserve high-frequency details without introducing excessive artifacts and without unnatural edge enhancement.

### 4.3. Generalization Experiment

To verify the generalization ability of the proposed model, 48 pairs of images are selected from the M3FD dataset [20] for testing. The model is not trained on the M3FD dataset and is directly used for testing. The M3FD dataset is a visible–infrared fusion dataset, containing 4200 pairs of images for fusion, detection, and fusion-based detection, as well as 300 pairs of images for independent scene fusion.

#### 4.3.1. Qualitative Comparison

As shown in Figure 8, first, in the results of RFN-Nest, the information on the signs is obscured in darkness and cannot be seen clearly (as shown in the green box in the first row of images and the red box in the second row of images). Second, the results of RFN-Nest and FusionGAN are generally blurrier than the original images (such as the tree areas in the first row of images). In the results of DIVFusion, LEFuse, and LENFusion, snowflake-like noise appears in the sky area of the first row of images, and the outline of the clouds is eroded (as shown in the red box in the first row of images). In the second row of images, the outline of distant buildings becomes incomplete compared with the original image. In addition, the results of these three methods all have varying degrees of color distortion, and Figure 8f even gives an overexposed feeling.

#### 4.3.2. Quantitative Analysis

Table 2 presents the average quality metrics of different fusion methods on the M3FD dataset. The method proposed in this study ranks first in the Qabf and ΔE metrics, second in the SF and CC metrics, and third in the Nabf, SCD and MS-SSIM metrics. As mentioned earlier, this indicates that the method proposed here can retain relatively rich texture details while avoiding the introduction of excessive artifacts, and it can better maintain the linear correlation, structure and color consistency between the fused image and the source images.

### 4.4. Ablation Experiment

To verify the effectiveness of each loss function and module in the method presented in this study, we conducted ablation experiments. To validate the effectiveness of the perceptual loss and the auxiliary intensity loss, we removed these two loss functions. To verify the effectiveness of the encoder in the FPN structure, we replaced it with the most basic convolutional layer, and the results are shown in Figure 9.

#### 4.4.1. Qualitative Comparison

When the perceptual loss $L_{per}$ is removed, the fused image exhibits severe color deviation and block artifacts. In fact, at this time, the RID-Net cannot accurately separate the reflectance component $I_{R}$. When the auxiliary intensity loss $L_{aux}$ is removed, the fused image fails to express sufficient infrared information (as shown in the green box in Figure 9d) and a lot of road texture information is lost (as shown in the red box in Figure 9d). When the encoder is replaced with a convolutional layer, it can be observed that the overall brightness of the image decreases.

#### 4.4.2. Quantitative Analysis

The quantitative results of the ablation experiment are shown in Table 3. The proposed method achieves better results in the CC, Qabf, SCD, and ΔE metrics. The SF is not optimal because areas with block artifacts often have higher spatial frequencies. The MS-SSIM is also not optimal because, without the auxiliary intensity loss $L_{aux}$, the fused image tends to be closer to either the visible image or the infrared image.

## 5. Discussion

Compared with other methods, the advantage of the one proposed here lies in its ability to retain a greater degree of consistency in structure and color between the original image and the fused image, and it can effectively suppress the image block effect and visual overexposure in some areas of the image. The limitations of this study lie in the fact that the encoder of the FPN structure has certain requirements for the size of the input image; in particular, it requires that the height and width of the image be multiples of 8. As such, images that do not meet these requirements need to undergo preprocessing. There are certain deficiencies in the feature extraction of infrared images, as the unique advantages of infrared images have not been separated and fully utilized. The feature fusion part adopts a simple concatenation rather than a deeper level of information fusion. In addition, the method of decomposing the illumination and reflection components is prone to overexposure in nighttime scenes including light sources, and the information expressed by the visible light image in the overexposed areas is insufficient. In this case, more infrared image information should be integrated to make up for this deficiency. Future considerations could include improving the feature fusion module to achieve deeper information integration between visible and infrared images; adding a light intensity adjustment module to solve the overexposure at the light source.

## 6. Conclusions

This study proposed a new fusion method to address the shortcomings of existing approaches for visible light and infrared image fusion in low-light scenarios. CDFFusion adopts a two-stage strategy, which mitigates visual overexposure, the image block effect, and color deviation. Specifically, we first trained a feature extraction network that can extract the features of the source image and separate the illumination component and reflection component of the visible light image. Then, we trained a fusion network that combines the proposed color mapping formula to achieve color-agnostic fusion of the infrared image and the enhanced visible light image. Qualitative and quantitative experimental results show that, compared with five existing methods, the proposed method has overall better results in both subjective and objective respects.

## Figures and Tables

**Figure 1 sensors-25-07337-f001:**
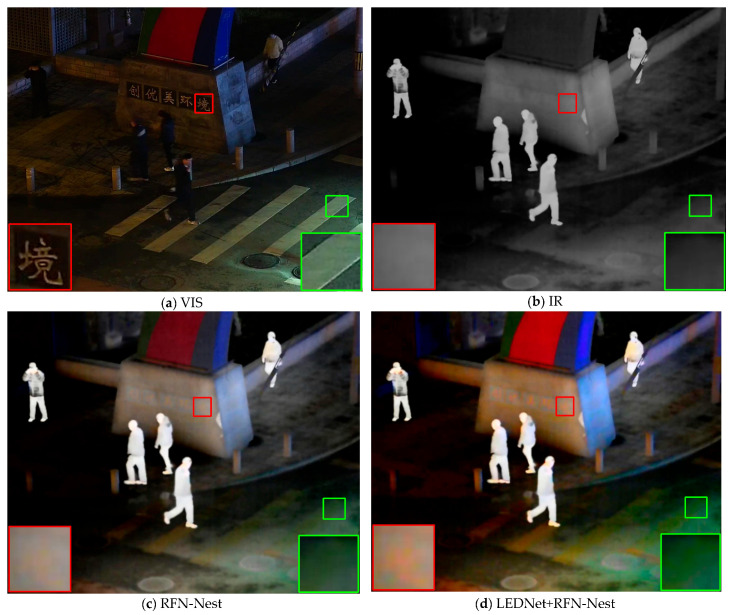
Fusion results of previous methods on the LLVIP dataset.

**Figure 2 sensors-25-07337-f002:**
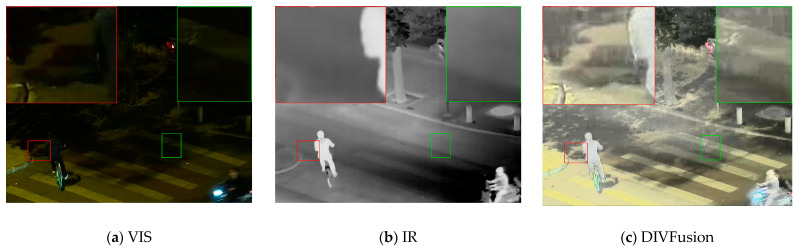
Fusion results of some recent methods and the proposed method on the LLVIP dataset.

**Figure 3 sensors-25-07337-f003:**
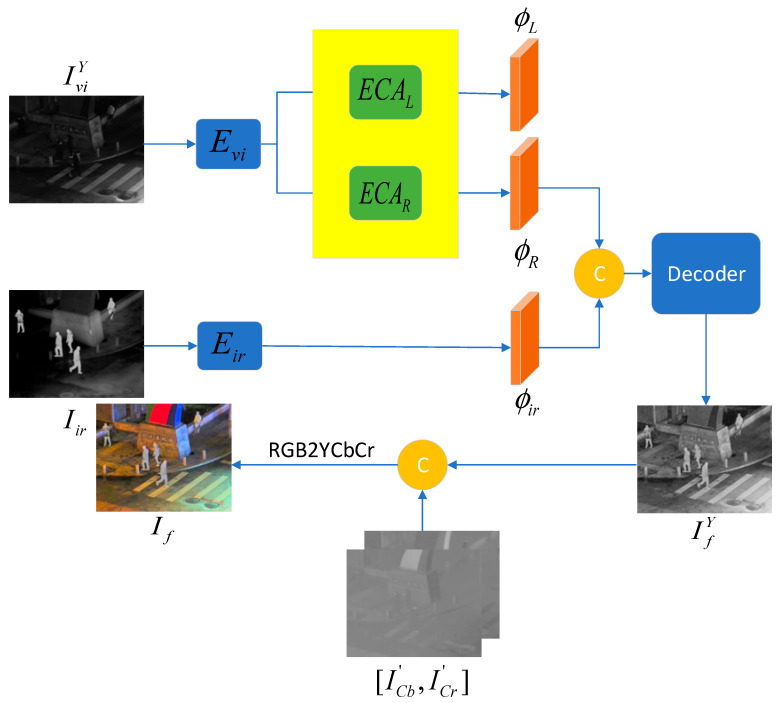
Overall framework of CDFFusion.

**Figure 4 sensors-25-07337-f004:**
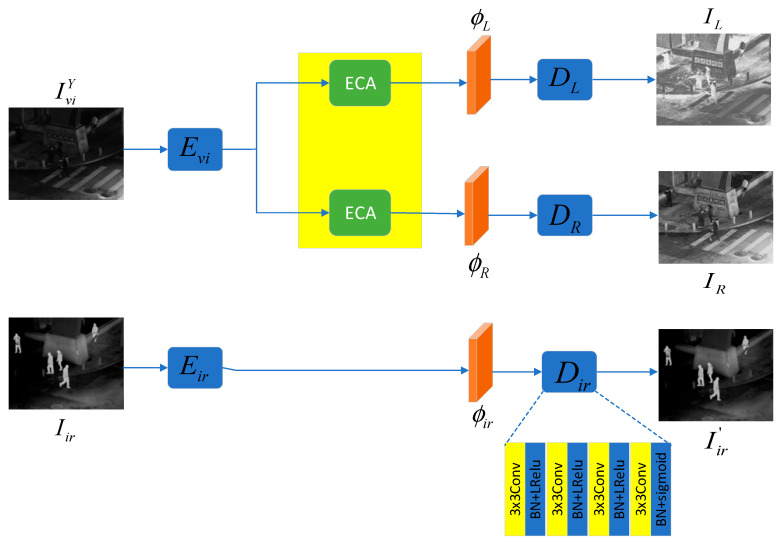
Structure of the RID-Net.

**Figure 5 sensors-25-07337-f005:**
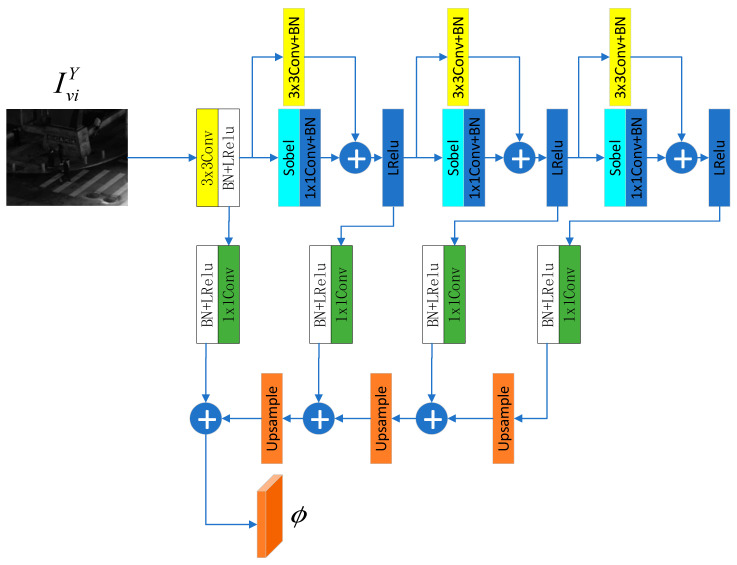
Structure of the encoder.

**Figure 6 sensors-25-07337-f006:**
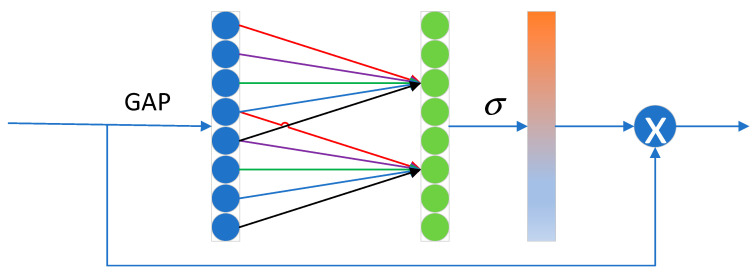
Structure of the ECA module.

**Figure 7 sensors-25-07337-f007:**
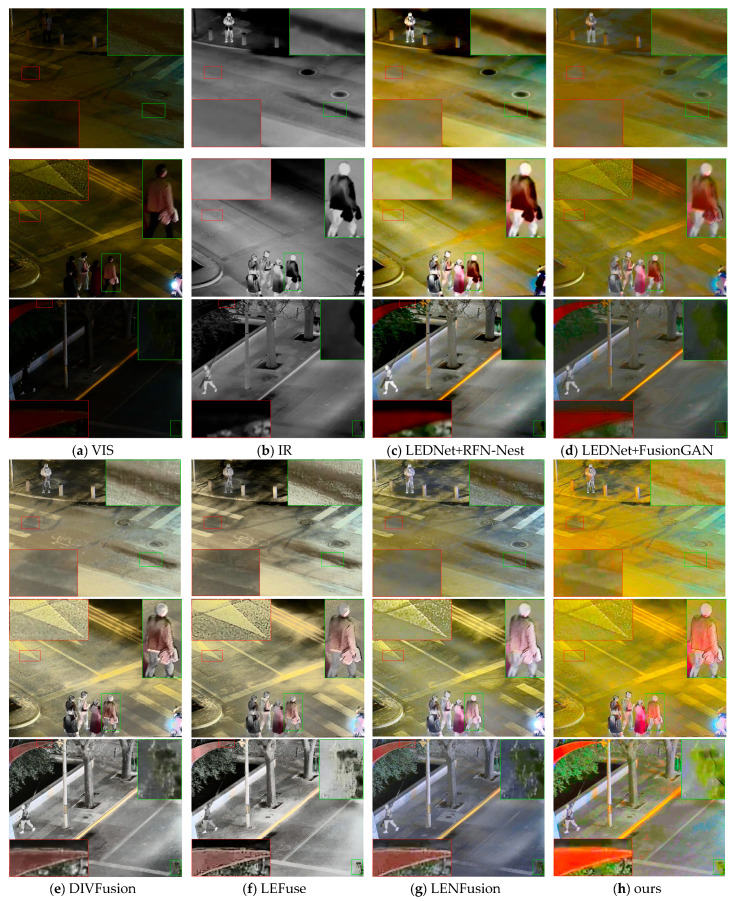
Experimental results of different methods on the LLVIP dataset.

**Figure 8 sensors-25-07337-f008:**
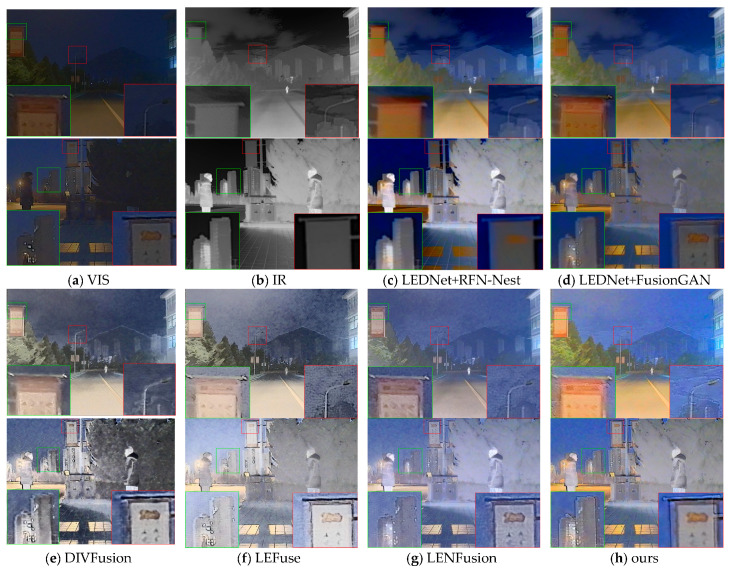
Experimental results of different methods on the M3FD dataset.

**Figure 9 sensors-25-07337-f009:**
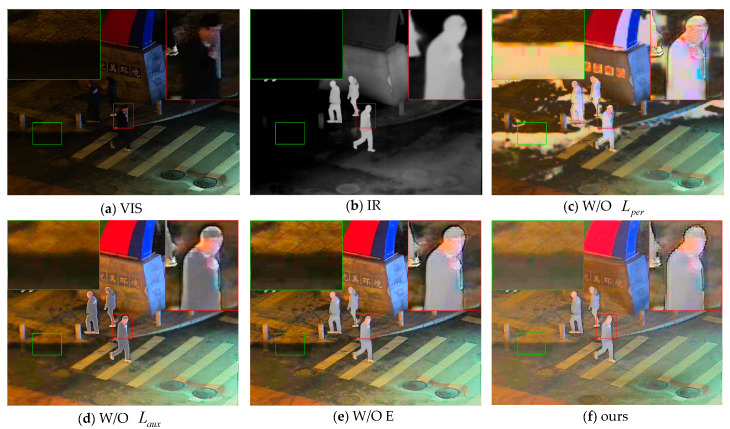
Comparison of results of the ablation experiment on perceptual loss $L_{per}$ and auxiliary intensity loss $L_{aux}$ based on the LLVIP dataset.

**Table 1 sensors-25-07337-t001:** Quantitative comparison between the proposed method and other methods on the LLVIP dataset. The best and second-best results are marked in bold and underlined, respectively.

	SF	CC	Nabf	Qabf	SCD	MS-SSIM	ΔE
RFN-Nest	5.0344	0.5824	**0.0099**	0.3332	1.0923	0.7730	0.2501
FusionGAN	6.7466	0.6479	0.0124	0.2788	0.9159	0.8188	0.1559
DIVFusion	14.7371	**0.6922**	0.1364	0.3823	**1.5373**	0.7973	0.3388
LEFuse	**24.0328**	0.6087	0.1856	0.3006	1.2262	0.7027	0.3141
LENFusion	21.4990	0.5928	0.1969	0.3534	1.0440	0.7236	0.3350
Ours	17.6531	0.6619	0.1075	**0.4279**	1.2760	**0.8335**	**0.0706**

**Table 2 sensors-25-07337-t002:** Quantitative comparison between the proposed method and other methods on the M3FD dataset. The best and second-best results are marked in bold and underlined, respectively.

	SF	CC	Nabf	Qabf	SCD	MS-SSIM	ΔE
RFN-Nest	4.1495	0.6260	**0.0039**	0.3133	0.9690	0.8565	0.1968
FusionGAN	6.2380	0.6579	0.0143	0.2770	0.7082	**0.8788**	0.1399
DIVFusion	14.4927	**0.7360**	0.1525	0.3684	**1.6140**	0.8276	0.0785
LEFuse	**20.6205**	0.6623	0.2416	0.2504	1.2501	0.7236	0.1190
LENFusion	14.6193	0.6640	0.1402	0.3637	0.9860	0.8011	0.0883
ours	16.4501	0.6879	0.1251	**0.3694**	1.0070	0.8416	**0.0269**

**Table 3 sensors-25-07337-t003:** Results of the ablation experiment based on the LLVIP dataset. The best results are marked in bold.

	SF	CC	Nabf	Qabf	SCD	MS-SSIM	ΔE
W/O $L_{per}$	**17.7592**	0.209	**0.0868**	0.4103	0.1861	0.6925	0.0750
W/O $L_{aux}$	15.3465	0.6607	0.0957	0.4169	1.242	**0.8699**	0.0838
W/O E	15.5647	0.5085	0.1080	0.2774	0.5027	0.7765	0.0855
ours	17.6531	**0.6619**	0.1075	**0.4279**	**1.276**	0.8335	**0.0706**

## Data Availability

The data presented in this study are openly available in [LLVIP] at https://openaccess.thecvf.com/content/ICCV2021W/RLQ/html/Jia_LLVIP_A_Visible-Infrared_Paired_Dataset_for_Low-Light_Vision_ICCVW_2021_paper.html (accessed on 11 July 2025), reference number [19].

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
