# Peer review of "Sensors2025, 25(23), 7337;https://doi.org/10.3390/s25237337"

_sensors, 2025, doi:10.3390/s25237337_

Round 1

Reviewer 1 Report

Comments and Suggestions for Authors

This paper addresses the important and pressing problem of fusion of infrared and visible images in low-light conditions. The authors propose a two-stage method, CDFFusion, that combines luminance enhancement and image fusion, simultaneously addressing overexposure, block artifacts, and color distortion. The authors propose a method that preserves natural color rendition while significantly improving the visibility of dark areas and integrating additional thermal information from infrared images. The approach is based on image separation into luminance and reflectance components, followed by color transformation. The method consists of two stages: first, an encoder is developed to extract deep details from visible and infrared images, respectively. Then, in conjunction with RetiNex theory, a decomposition network is designed at the detail level to separate the luminance and reflectance components of the visible-light image. The proposed formula is then used to process the Cb and Cr components of the original visible-light image. The reflectance and infrared component features are combined and fed into a fusion network to produce the Y component of the fused image. This component is finally combined with the processed Cb' and Cr' components of the visible image to produce the final fused image. This paper addresses current challenges in computer vision and has practical implications for applications such as autonomous driving, video surveillance, and robotics. While the fusion problem is not new, the proposed approach contains original elements. The authors address a specific gap: systematic color additions in existing joint enhancement and fusion methods.

Compared to other published materials, this work makes a significant contribution: a rigorous mathematical justification for chroma preservation through synchronous scaling of Y, Cb, and Cr components; a two-stage architecture with individual enhancement and fusion at the feature level; a combination of Retinex decomposition with ECA attention to separate illumination and reflectance components; explicit chroma processing via Cb/Cr feature transformation instead of implicit blocking via loss functions; and the elimination of systematic artifacts inherent in modern methods.

However, the paper contains a number of comments that require correction before publication.
- Some notations are unclear (e.g., I_vi^Cb and I_vi^Cb' in formula (11)).
- Tables 2 and 3 lack references to the methods used for comparison.
- The equipment used for the experiments is not specified.

The proposed method is of scientific interest and solves an important practical problem. However, these issues need to be addressed before publication. After revision, the paper may be of significant value to the scientific community. 

Author Response

Thank you very much for taking the time to review this manuscript. Please find the detailed responses below and the corresponding revisions. My changes to the manuscript are highlighted in red.

Reviewer 2 Report

Comments and Suggestions for Authors

The manuscript proposes CDFFusion, a network designed to jointly enhance and fuse visible and infrared images under low-light conditions while avoiding color deviation. Experiments are conducted on LLVIP and M3FD datasets and compared with several state-of-the-art (SOTA) model.

The topic fits well within journal's scope. The problem statement is relevant, and the color-deviation issue is practically meaningful. However, the novelty is not fully validated experimentally, and the presentation is overly mathematical, reducing readability. Furthermore, the experimental section lacks diversity.
Overall, the work shows potential but requires significant revision to reach publication quality.

  1. The abstract is overly long and descriptive. It should follow a concise structure: backgrounds → method → results → conclusion.
  2. Please include numerical performance highlights to quantify the improvement in abstract.
  3. Throughout the entire manuscript, the number of references is insufficient to properly support the technical descriptions, methodological choices, and experimental discussions. Please strengthen the paper by adding more relevant and up-to-date citations across all sections to provide adequate theoretical and empirical context.
  4. It is recommended to restructure the manuscript following the IMRAD format (Introduction, Methods, Results, and Discussion) to enhance logical flow, readability, and alignment with standard scientific writing conventions.
  5. The motivation for ECA modules is unclear. Why was ECA chosen over SE or CBAM? Add a brief justification.
  6. The dataset usage (240 training pairs, 50 test pairs from LLVIP) is rather small. Please discuss potential overfitting and whether data augmentation or pretraining was applied.
  7. Provide details about experiments, including hardware, training and inference time as these are common requirements in Sensors papers.
  8. The proposed method performs best on Qabf and MS-SSIM but not on other metrics. Discuss these trade-offs and possible reasons.
  9. Provide statistical significance (standard deviation or paired t-test) over multiple samples or runs (in Tables).
  10. Include color difference metrics to substantiate the “no color deviation” claim.
  11. The ablation experiment is too limited.
  12. The Discussion and Conclusion sections are too weak. In general, the Discussion section should include a concise literature comparison, both quantitative and qualitative analyses comparing the proposed method with existing approaches, as well as a clear description of limitations and future research directions.
  13. Additionally, the Conclusion section should be separated and written independently to summarize the main findings and contributions of the study.
Comments on the Quality of English Language

The overall English writing is understandable but requires refinement for academic style and readability. Please revise the manuscript to ensure concise, objective, and formal scientific language, avoiding exaggerated expressions (e.g., “no color deviation at all”) and improving sentence flow and consistency in terminology.

Author Response

(The authors gave the same response as above.)

Reviewer 3 Report

Comments and Suggestions for Authors

The paper proposes a visible light and infrared image fusion method, CDFFusion, for low-light scenarios. However, this paper has limited innovation, and there are obvious issues with its formatting and expression. It requires careful revision before publication. The specific issues are as follows:

1) The paper does not include ablation experiments about new modules, making it difficult to demonstrate the superiority of the work presented.

2) The network structure is disorganized, and the innovation is insufficient. 

First, the network uses conventional convolution for feature extraction and only relies on concatenation to achieve relevant fusion, resulting in insufficient innovation compared with existing methods. Second, the decoder in Figure 4 of this paper only includes operations such as convolution, which is similar to the feature extraction function of the encoder rather than a decoder used in the upsampling process; thus, its effectiveness is difficult to guarantee. Finally, the encoder in Figure 5 adopts a Unet network structure, which actually contains both an encoder and a decoder—this is an incorrect description. It is suggested that the authors further understand the function of each structure and revise the paper accordingly. 3) There are multiple errors in the first-line indentation of the paper; please further check and correct them. Additionally, please review the English grammar throughout the entire paper.

Author Response

(The authors gave the same response as above.)

Round 2

Reviewer 2 Report

Comments and Suggestions for Authors

The authors have significantly improved the manuscript, and most of the major concerns raised in the previous review have been addressed.

One remaining aspect that would benefit from further improvement is the Discussion section.
Although the authors have added a dedicated Discussion, the current version still reads relatively briefly and does not fully analyze:

  • How the proposed method compares to existing approaches beyond reporting quantitative metrics.

  • Clear articulation of limitations.

  • Implications and directions for future research.

Author Response

(The authors gave the same response as above.)

Reviewer 3 Report

Comments and Suggestions for Authors

The author has addressed my concerns, and this paper is acceptable for publication.

Author Response

Thank you for your understanding and acceptance.
